# PointESS: Efficient 3D Point Cloud Modeling through Enhanced State Space Networks and Fine-Tuning Strategies

## Abstract

Recent progress in point cloud modeling highlights the need for efficient pretraining strategies and task adaptability. We introduce *PointESS*, the first framework to leverage the *Mamba2* architecture for point cloud representation learning. PointESS balances structural modeling with downstream flexibility and incorporates two key components to enhance local structural awareness and fine-tuning performance: Gate Adapter, which dynamically integrates pretrained and task-specific features for improved transferability, and KH-Norm (KNN-Hybrid Normalization), which embeds geometric priors into normalization and pooling to capture fine-grained spatial details. Extensive experiments on ShapeNet55, ModelNet40, and ScanObjectNN demonstrate that PointESS achieves competitive or state-of-the-art results with substantially fewer parameters, highlighting its effectiveness in both generalization and efficiency.

**Keywords**: Point Cloud, Mamba2, KH-Norm, Gate Adapter, 3D Object Classification

## 1 Introduction

In recent years, advances in 3D sensing technologies (e.g., LiDAR, RGB-D cameras, 3D scanners) have enabled large-scale acquisition of point cloud data, fueling applications in transportation, virtual reality (VR), industrial inspection, and BIM. The global market for 3D point cloud processing software was USD 1.15 billion in 2024 and is projected to reach USD 3.06 billion by 2031, reflecting its growing industrial value.

Unlike structured data such as images or audio, point clouds are sparse, irregular, and unordered. Lacking explicit topological connections and fixed spatial order, they are incompatible with traditional deep learning models—particularly CNNs designed for regular grids. Additionally, non-uniform point distributions and variable point counts further complicate geometric modeling and feature extraction. To tackle these challenges, researchers have pursued two main strategies. One transforms point clouds into structured formats, such as voxels or depth maps, allowing conventional CNNs to be applied. The other directly processes unordered point sets with architectures like *PointNet* Qi et al. (2017a), *PointNet++* Qi et al. (2017b), and *DGCNN* Wang et al. (2019). These models capture geometric relationships within local neighborhoods. While effective, these methods often incur high model complexity, heavy training costs, and slow inference, motivating the development of more efficient architectures and fine-tuning strategies for 3D point clouds.

To improve efficiency in point cloud processing, researchers have explored architectures that capture long-range dependencies and global context. Transformer-based models, which have revolutionized NLP and image understanding, have been successfully adapted for 3D point clouds. For instance, *Point Transformer* Zhao et al. (2021) incorporate self-attention mechanisms to model the relationships between each point and its local neighborhood, thereby enhancing the recognition of local geometric structures. Methods like *Point-BERT* Yu et al. (2022) and *PointMAE* Pang et al. (2022) extend the pretraining paradigms of *BERT* Devlin et al. (2019) and *MAE* He et al. (2021) by applying self-supervised learning through masked point cloud patches, significantly improving performance in downstream tasks such as classification and segmentation. These approaches highlight the potential of Transformer encoders for learning rich 3D point cloud representations. However,

the self-attention mechanism in Transformers scales quadratically with sequence length ($\mathcal{O}(L^2)$), making dense point clouds or long sequences computationally expensive and memory-intensive. To overcome this bottleneck, researchers have explored architectures that preserve long-range modeling while achieving linear efficiency and better scalability. In this context, *State Space Models (SSMs)* have emerged as a promising alternative.

To address the efficiency limits of Transformers in long-sequence modeling, SSMs have gained renewed interest. Originally developed for signal processing and control, SSMs model long-term dependencies via a recurrent formulation and can process sequences of length $L$ with linear complexity $\mathcal{O}(L)$, unlike Transformers' quadratic scaling, making them more efficient and scalable. Among recent advances, the *Mamba* Gu & Dao (2024) family exemplifies SSM-based architectures, introducing selective state space mechanisms with data-dependent gating and parallel-friendly design. This preserves linear-time efficiency while enhancing learning capacity and flexibility. With strong modeling power and high efficiency, *Mamba* achieves performance comparable to Transformers in language and image tasks, inspiring its extension to 3D point cloud processing.

Recent studies have explored adapting the *Mamba* architecture for 3D scene modeling to address the unique properties of point clouds. *PointMamba* Liang et al. (2024b) is among the first to apply selective state space mechanisms to point cloud processing. By replacing complex attention modules with a streamlined design and leveraging point ordering and spatial encoding, *PointMamba* delivers accurate and efficient classification and segmentation. Its compact architecture achieves results comparable to large Transformers, demonstrating the promise of SSM-based methods for 3D tasks.

*Mamba3D* Han et al. (2024) extends this approach with a more sophisticated point cloud architecture, introducing modular *Mamba* blocks designed for the spatial and hierarchical nature of 3D data. By incorporating multi-scale feature extraction and refined positional encodings, it improves the capture of geometric shapes and fine-grained structures. *Mamba3D* achieves strong results on benchmarks like ModelNet40 and ShapeNet, demonstrating the effectiveness of state space models for point cloud learning.

Despite SSMs' success in language and vision tasks, few fine-tuning strategies or architectures address the unique geometry of 3D point clouds. To fill this gap, we propose *PointESS*, a lightweight, modular SSM-based architecture for efficient point cloud modeling and fine-tuning. PointESS delivers competitive performance on standard benchmarks while substantially reducing parameters and inference cost. The main contributions of this work are summarized as follows:

- We are the first to apply the *Mamba2* architecture to 3D point clouds, demonstrating its effectiveness for sparse spatial data.
- We introduce the Gate Adapter, a lightweight, fully fine-tuned module that enables task-specific adaptation while maintaining structural simplicity, unlike typical PEFT methods.
- We propose the KH-Norm, a novel normalization technique that combines coordinate and feature information to improve geometric regularization and local structure learning.
- PointESS achieves strong and stable classification performance with a smaller model size and lower inference cost compared to existing methods.

## 2 STATE SPACE MODELS AND MAMBA ARCHITECTURE

State Space Models (SSMs) originate from continuous-time physical and control systems. At their core, SSMs update a hidden state $h_t$ recursively and compute an output $y_t$ based on that state. For discrete input scenarios—such as text, images, or point clouds—the continuous formulation of SSMs can be discretized as:

$$h_t = Ah_{t-1} + Bx_t, \quad y_t = Ch_t$$

where $A$, $B$, and $C$ are the state transition matrix, input control matrix, and output matrix, respectively. In classical design, these parameters are fixed, making the system a *Linear Time-Invariant (LTI)* system. One major advantage of LTI systems is their ability to transform the recursive relationship into a global convolution form, allowing for parallelized inference and efficient sequence modeling. Specifically, for a sequence of length $L$, the input $x$ can be processed using a precomputed convolution kernel:

$$K = [CB, CAB, \ldots, CA^{L-1}B], \quad y = x * K$$

To further improve modeling capacity, *S4* introduced a structured parameterization and incorporated the *HiPPO (Highly Parallel Polynomial Operator)* framework, enabling SSMs to handle longer sequences without sacrificing speed—establishing S4 as a representative architecture for efficient long-sequence modeling. However, the fixed parameters $A$, $B$, and $C$ in LTI systems limit their ability to adapt information flow based on input content, lacking selective control over contextual information. To address this limitation, *Mamba* introduces a selective state space design in which the parameters are dynamically generated from the input:

$$h_t = \tilde{A}(x_t)h_{t-1} + \tilde{B}(x_t)x_t, \quad y_t = \tilde{C}(x_t)h_t$$

Here, each parameter is a projection of the input $x_t$, making the system *input-dependent*. This design enables the model to learn different information propagation rules at each time step, enhancing its capacity for context-aware modeling.

However, this dynamic parameterization breaks the parallelism of the original LTI structure, preventing the use of precomputed convolutions for fast inference and introducing efficiency bottlenecks. To resolve this, *Mamba2* was designed with the goal: "retain selectivity, but make it faster and simpler." Its core principle is the *Structured State Space Duality (SSD)*, which reveals that under certain conditions, state space models and self-attention are structurally equivalent in terms of information modeling. Based on this insight, Mamba2 integrates the three separate projection modules into a single linear transformation layer, simplifying the parameter generation process while preserving input-dependency and achieving low-latency inference. This greatly narrows the performance gap between SSMs and Transformer-based architectures. Overall, *Mamba2* combines the linear efficiency of SSMs with selective control mechanisms, offering a compelling and efficient alternative for long-sequence modeling.

## 3 PROPOSED METHOD

### 3.1 PRELIMINARY

Raw point clouds often contain thousands to millions of points, making direct processing inefficient. To address this, most methods downsample to representative subsets. A common approach is to combine *Farthest Point Sampling* (FPS) with *K-Nearest Neighbors* (KNN). We first apply FPS to select $n$ uniformly distributed center points $C \in \mathbb{R}^{n \times 3}$. For each center $c_i \in \mathbb{R}^{1 \times 3}$, KNN retrieves $k$ neighboring points, forming a local patch $G_i \in \mathbb{R}^{k \times 3}$. This yields a structured set of local patches $\mathrm{G} = \{ G_i \}_{i=1}^{n} \in \mathbb{R}^{n \times k \times 3}$, which preserves spatial consistency while reducing computational cost.

Each patch $G_i \in \mathbb{R}^{k \times 3}$ is then mapped by a point-wise MLP from $\mathbb{R}^3 \to \mathbb{R}^d$, producing per-point features $F_i \in \mathbb{R}^{k \times d}$. A max-pooling operation aggregates the $k$ neighboring points into a fixed-length vector $f_i \in \mathbb{R}^d$. Collectively, these $n$ vectors form a sequence $F \in \mathbb{R}^{n \times d}$, which is then fed into the Mamba2 encoder to model sequential dependencies among patches for higher-level representation learning.

### 3.2 POINTESS WORKFLOW

We propose a novel architecture *PointESS*, designed to balance effective point cloud representation with adaptability to downstream tasks. As illustrated in Figure 1, PointESS follows a two-stage pipeline: pre-training and fine-tuning, with two key components: the *Gate Adapter* and *KH-Norm* in the encoder.

In pre-training, an autoencoder framework learns semantic representations of 3D point clouds. The input undergoes FPS downsampling and KNN grouping to form local regions, which are processed by a patch embedding module to extract geometric features. Random masking of patches, inspired by masked language modeling, guides the model to capture global structure. The encoder consists of a Mamba2-based sequential modeling backbone, augmented with our proposed *KH-Norm* (KNN-Hybrid Normalization), which enhances sensitivity to local geometric variations. Positional

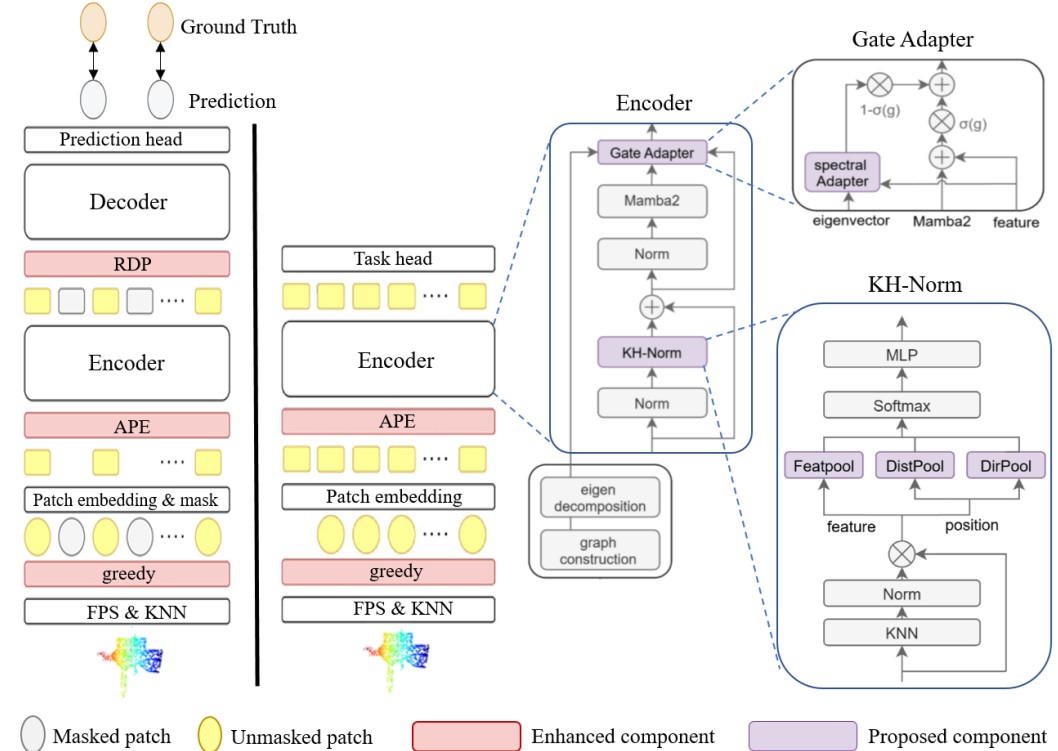

Figure 1: Illustration of the PointESS architecture. Left: pre-training stage (with masked reconstruction). Right: fine-tuning stage (connected to task head). Red modules indicate enhanced components, while purple modules denote our proposed contributions.

information is incorporated via absolute position encoding (APE) and a relative direction prompt (RDP) before masked patch reconstruction by the decoder.

During fine-tuning, the decoder is removed, and encoder features are fed into a classification head for tasks such as object classification and semantic segmentation. Lightweight, task-aware *Gate Adapter* modules within the Mamba2 blocks allow efficient feature modulation across tasks, improving adaptability and transferability without modifying the architecture.

To further improve the model's ability to capture geometric and sequential relationships, we incorporate three auxiliary strategies:

- *Greedy Sorting:* Since point clouds are unordered, we impose a meaningful sequence by starting from a random patch and iteratively selecting the nearest patch as the next element. This generates a spatially coherent patch sequence that enhances local geometric context modeling.

- *APE (Absolute Positional Encoding):* We add absolute positional information to each input patch to help the model distinguish spatial locations and enhance structural awareness. The APE is computed from the geometric center of each patch and projected via a linear layer before being added to the patch features.

- *RDP (Relative Directional Prompt):* To provide spatial cues while maintaining uncertainty during pre-training, we calculate, for each unmasked patch, a normalized directional vector relative to its neighboring patch :

$$\mathbf{d}_i = \frac{C_{i+1}^s - C_i^s}{\|C_{i+1}^s - C_i^s\|_2}$$

where $C_i^s \in \mathbb{R}^3$ denotes the 3D coordinates of the $i$-th patch center in the sorted sequence. This vector guides the decoder in capturing local geometric trends without leaking absolute position information.

### 3.2.1 Encoder and Decoder

The proposed PointESS architecture is built upon Mamba2 and adopts a modular encoder–decoder pipeline to accommodate both pre-training and downstream task requirements. During the pre-training phase, we employ an autoencoder framework to perform masked reconstruction, enabling the model to gain a holistic understanding of 3D structures. In the fine-tuning phase, the decoder is removed, and only the encoder with a task head is retained for efficient task transfer.

The encoder is composed of stacked Mamba2 blocks, into which we integrate our proposed KH-Norm and Gate Adapter modules. These components enhance the model's sensitivity to local geometric variations and improve adaptability to downstream tasks. Each Mamba2 block takes the output of the previous encoder layer as input, enabling hierarchical representation learning. By leveraging the state space mechanism, each block is able to capture long-range semantic dependencies and effectively propagate information. Compared to transformer-based encoders, Mamba2 offers improved memory and runtime efficiency due to its linear-time characteristics, making it particularly suitable for modeling ordered point cloud patches.

The decoder is only activated during the pre-training phase. It is designed as a lightweight module consisting of several MLP layers and position-guided units that convert semantic features from the encoder into reconstructed point cloud outputs. Since the goal is masked reconstruction, only unmasked patch features are fed into the decoder. To assist reconstruction, we apply Relative Directional Prompt (RDP) to provide contextual directional cues. The decoder predicts the coordinates of masked patches, which are then supervised using the original ground-truth point positions by minimizing geometric losses such as Chamfer Distance.

Overall, the PointESS encoder exhibits strong semantic learning capacity and modular flexibility, supporting both structural understanding in pre-training and efficient feature adaptation in fine-tuning. The decoder complements this by helping the model grasp latent geometric structures, thereby enhancing pre-training effectiveness.

### 3.2.2 Gate Adapter: A Spectrally-Guided Dynamic Fine-Tuning Module

Recent work such as PointGST Liang et al. (2024a) integrates graph spectral analysis into parameter-efficient fine-tuning (PEFT) for point clouds. Using Graph Fourier Transform (GFT), it adjusts frequency component weights in a lightweight manner to enhance adaptability. A point cloud is represented as a graph $G = \{V, E, W\}$, with vertices $V$ (corresponding to a point patch), edges $E$, and adjacency weight matrix $W \in \mathbb{R}^{n \times n}$. From $W$, the graph Laplacian is defined as $L = D - W$, where $D \in \mathbb{R}^{n \times n}$ is the diagonal degree matrix with entries $d_{ii} = \sum_{j=1}^{n} w_{ij}$ representing the degree of each vertex. Eigendecomposition of $L$ yields $L = U\Lambda U^{\top}$, with $U \in \mathbb{R}^{n \times n}$ as spectral basis, and $\Lambda \in \mathbb{R}^{n \times n}$ as eigenvalues, enabling spatial-frequency domain transformation:

$$\hat{z} = U^{\top} z, \quad z = U\hat{z}$$

Unlike PointGST, which relies on a frozen backbone under PEFT paradigm, we target fully fine-tuned architectures where all parameters are updated. To manage this, we propose the *Gate Adapter*, a module that regulates the fusion of pretrained and task-specific features. It introduces a trainable gating parameter $g$ to control the fusion:

$$\hat{z} = \sigma(g) \cdot z_{\text{pretrained}} + (1 - \sigma(g)) \cdot z_{\text{adapted}}$$

where $\sigma(\cdot)$ is the sigmoid function, constraining $g$ to $[0, 1]$. This mechanism enables adaptive balancing of pretrained and task-specific features, preserves semantic stability during full fine-tuning, and decouples frequency pathways to avoid single-stream dominance.

### 3.2.3 KH-Norm: KNN-Hybrid Normalization

Traditional normalization methods in visual models, such as LayerNorm and InstanceNorm, operate across channels or batch dimensions but ignore the geometric structure of the data. This limitation is particularly problematic for irregular, topology-free point clouds. To address this, we propose *KNN-Hybrid Normalization (KH-Norm)*, a geometry-aware normalization method that integrates feature

standardization with spatially weighted aggregation. Inspired by the LNP block in Mamba3D Han et al. (2024), KH-Norm follows a two-stage paradigm of diffusion and aggregation.

*Step 1: Geometry-Guided Feature Normalization*

This is the diffusion phase, where each patch serves as a center patch and diffuses its features to its neighbors for normalization. For a center patch $F_C \in \mathbb{R}^{1 \times d}$ and its $K$ nearest neighboring patches $F_K \in \mathbb{R}^{K \times d}$, we compute the standardized difference:

$$\tilde{F}_K = \frac{F_K - F_C}{\text{Var}(F_K - F_C) + \epsilon}$$

where $\text{Var}(\cdot)$ is computed across the $K$ neighbors for each feature channel. The normalized neighbors are concatenated with the center patch feature and transformed via learnable affine parameters $\gamma \in \mathbb{R}^{1 \times d}$ and $\beta \in \mathbb{R}^{1 \times d}$:

$$\hat{F}_K = [\tilde{F}_K \oplus F_C] \cdot \gamma + \beta$$

where $\oplus$ denotes channel-wise concatenation. This step emphasizes relative geometric variations and provides the basis for spatially aware aggregation.

*Step 2: Multi-Geometry-Aware Pooling*

This is the aggregation phase, where the normalized features are pooled using three complementary spatial strategies:

*1. FeatPool (Feature-Sensitive Pooling)*: Softmax-style attention over feature magnitudes:

$$\text{FeatPool}(x) = \sum_{k=1}^{K} \frac{e^{x_k}}{\sum_{j=1}^{K} e^{x_j}} \cdot x_k$$

where $x_k \in \mathbb{R}^d$ denotes the feature vector of the $k$-th neighbor within a local patch of $K$ points, and the exponential function is applied element-wise to each feature dimension. Features with larger activations receive higher weights, enabling the model to emphasize salient local responses.

*2. DistPool (Distance-Aware Pooling)*: Weights inversely proportional to distance from the center:

$$w_k = \frac{1/(\|p_k\| + \epsilon)}{\sum_{j=1}^{K} 1/(\|p_j\| + \epsilon)}, \quad \text{DistPool}(x, p) = \sum_{k=1}^{K} w_k \cdot x_k$$

where $p_k \in \mathbb{R}^3$ represents the relative position vector from the patch center to its $k$-th neighbor (i.e., $p_k = x_k^{\text{pos}} - x_{\text{center}}^{\text{pos}}$), and $\|p_k\|$ is the Euclidean distance. This pooling favors spatially proximal features.

*3. DirPool (Direction-Aware Pooling)*: Learnable attention on unit direction vectors $d_k = \frac{p_k}{\|p_k\|}$:

$$w_k = \sigma(\phi(d_k)), \quad w_k^{\text{norm}} = \frac{w_k}{\sum_{j=1}^{K} w_j}, \quad \text{DirPool}(x, d) = \sum_{k=1}^{K} w_k^{\text{norm}} \cdot x_k$$

where $d_k \in \mathbb{R}^3$ is the unit direction vector from the patch center to its $k$-th neighbor, encoding geometric orientation. The function $\phi : \mathbb{R}^3 \to \mathbb{R}$ is a learnable MLP that maps directional vectors to scalar importance values, and $\sigma(\cdot)$ denotes the sigmoid function to constrain weights into the range $[0, 1]$. The weights $w_k^{\text{norm}}$ are then normalized across all neighbors to form a probability distribution. This pooling enables the model to attend to specific spatial directions based on geometric alignment.

These pooled outputs are concatenated and passed through a learnable mapping layer to produce the final output of KH-Norm. By explicitly combining diffusion and aggregation in a geometry-aware fashion, KH-Norm provides a more discriminative and spatially consistent normalization framework tailored for point cloud representation learning.

### 3.2.4 Loss Function

To balance semantic understanding and task adaptability, we employ different loss functions for the pre-training and fine-tuning stages of PointESS.

In pre-training, the goal is to reconstruct masked point cloud patches from visible ones using a masked autoencoder. The model observes only the visible patch features and predicts the geometry of the masked patches. We use the *Chamfer Distance (CD)* as the reconstruction loss, which measures the geometric discrepancy between the predicted and ground truth point sets:

$$\mathcal{L}_{CD}(P, Q) = \frac{1}{|P|} \sum_{p \in P} \min_{q \in Q} \|p - q\|_2^2 + \frac{1}{|Q|} \sum_{q \in Q} \min_{p \in P} \|q - p\|_2^2$$

where $P$ denotes predicted points and $Q$ the ground truth. This metric encourages accurate reconstruction of local shapes and contours while being permutation-invariant, making it well-suited for point cloud data.

In fine-tuning, the pre-trained encoder is adapted to downstream tasks such as classification or segmentation by attaching a task-specific MLP head. We adopt the standard *Cross-Entropy Loss*, which penalizes the divergence between the predicted probability distribution and the ground truth label:

$$\mathcal{L}_{CE} = -\sum_{i=1}^{C} y_i \log(\hat{y}_i)$$

where $C$ is the number of classes, $y_i$ the one-hot ground truth, and $\hat{y}_i$ the predicted probability for class $i$. This loss guides the model to learn discriminative features tailored to each class.

## 4 Experiments

### 4.1 Experiment Setup

All experiments are conducted on a single NVIDIA RTX TITAN GPU using PyTorch. We use the AdamW optimizer (lr=0.001, weight decay=0.05) with cosine decay scheduling, 10 warm-up epochs, 300 total epochs, batch size 64, and step-wise gradient updates. For pre-training, the model is trained on ShapeNet-55 ( 50k CAD models, 55 categories), where each shape is represented as a 1024-point cloud. We sample 64 patch centers using *FPS* and group 32 neighbors with *KNN*, yielding 64 patches per shape that capture local geometry. Training follows an autoencoding paradigm with masked point reconstruction, optimized by the Chamfer-L2 loss. The encoder is based on our *Mamba2* architecture with *KH-Norm*. Although *Gate Adapter* is part of the architecture, it remains inactive to ensure task-agnostic learning. During fine-tuning, the decoder is discarded for efficiency, and the *Gate Adapter* is activated to enable adaptive feature modulation for downstream tasks.

### 4.2 Datasets

We evaluate performance on two complementary benchmarks that span both clean synthetic data and noisy real-world scans. *ModelNet40* is a synthetic dataset of about 12,000 CAD models across 40 categories, widely used for point cloud recognition, with 9,843 samples for training and 2,468 for testing. Following standard protocol, we uniformly sample 1024 points per model. *ScanObjectNN* is a real-world dataset with 15 categories featuring noisy, occluded, and deformed point clouds. It includes three subsets—OBJ_BG, OBJ_ONLY, and PB_T50_RS—that capture varying levels of preprocessing and difficulty, making it a critical testbed for robustness.

### 4.3 Classification Results

We present a comprehensive comparison of classification results in Table 1, reporting accuracy, parameter count, and FLOPs across a broad set of point cloud models. Experiments are conducted on ModelNet40 and ScanObjectNN, benchmarked against representative methods including Point-MAE, ACT, ReCon, PointGPT, and PointMamba, spanning diverse design paradigms and scales.

On ModelNet40, our method achieves 95.5% classification accuracy, surpassing all publicly reported methods while using only 16.8M parameters and 4.1 GFLOPs. This efficiency contrasts sharply with larger architectures such as ReCon++L (657.2M parameters, 94.8%) and PointGPT-L (360.5M parameters, 94.7%). Unlike prior works that depend on deep backbones, heavy augmentations, or external supervision signals, our method reaches state-of-the-art performance through architectural innovations like KH-Norm and Gate Adapter alone. Lightweight pre-training and fine-tuning further enhance the deployment feasibility of our approach.

On ScanObjectNN, which presents real-world challenges with occlusions, noise, and background clutter, our model achieves 96.04%, 95.00%, and 95.10% accuracy on the OBJ_BG, OBJ_ONLY, and PB_T50_RS subsets, outperforming most mainstream methods. Compared to Mamba-based methods such as PointMamba (89.31%) and Mamba3D (93.34%), we achieve gains of 1.76% and 1.10% on PB_T50_RS, and deliver an 8.7% improvement over lightweight Point-M2AE (86.43%). Although ReCon++L slightly surpasses us on PB_T50_RS (95.25% vs. 95.10%), it requires 657M parameters, which is nearly 40× larger than our model. Notably, we outperform ReCon++L on ModelNet40 (95.5% vs. 94.8%). Overall, our method offers a strong balance of accuracy, efficiency, and robustness across both synthetic and real-world datasets.

Table 1: Classification results on the ScanObjectNN and ModelNet40 datasets

| Method | Reference | #Params ↓ | #Flops ↓ | ScanObjectNN | | | ModelNet40 |
| --- | --- | --- | --- | --- | --- | --- | --- |
| | | | | OBJ_BG ↑ | OBJ_ONLY ↑ | PB_T50_RS ↑ | 1k P ↑ |
| PointBERT | CVPR 22 | 22.1M | 4.8G | 87.43 | 88.12 | 83.07 | 93.2 |
| MaskPoint | ECCV 22 | 22.1M | 4.8G | 89.30 | 88.10 | 84.30 | 93.8 |
| PointMAE | ECCV 22 | 22.1M | 4.8G | 90.02 | 88.29 | 85.18 | 93.8 |
| Point-M2AE | NeurIPS 22 | 15.3M | 3.6G | 91.22 | 88.81 | 86.43 | 94.0 |
| ACT | ICLR 23 | 22.1M | 4.8G | 93.29 | 91.91 | 88.21 | 93.7 |
| RECON | ICML 23 | 43.6M | – | 95.35 | 93.80 | 91.26 | 94.5 |
| PointGPT-L | NeurIPS 23 | 360.5M | 67.7G | 97.2 | 96.6 | 93.4 | 94.7 |
| Point-FEMAE | AAAI 24 | 27.4M | – | 95.18 | 93.29 | 90.22 | 94.5 |
| RECON++L | ECCV 24 | 657.2M | – | 98.80 | 97.59 | 95.25 | 94.8 |
| PointMamba | NeurIPS 24 | 12.3M | 3.1G | 94.32 | 92.60 | 89.31 | 93.6 |
| Mamba3D | ACM MM24 | 16.9M | 3.9G | 95.18 | 92.60 | 93.34 | 95.1 |
| Ours(Mamba) | – | 17.3M | 3.9G | 95.86 | 94.83 | 93.99 | 95.4 |
| Ours(Mamba2) | – | 16.8M | 4.1G | 96.04 | 95.35 | 95.10 | 95.5 |

## 4.4 ABLATION STUDY

### 4.4.1 PATCH SORTING STRATEGY

To examine the effect of patch ordering on model performance, Table 2 compares random ordering, Morton (Z-order) curve, and our proposed greedy sorting on ModelNet40. The greedy strategy achieves the highest accuracy (95.46%), outperforming random (94.77%) and Morton (94.93%), highlighting the importance of patch ordering for sequence-based models like Mamba2. By leveraging geometric proximity, greedy sorting produces semantically coherent sequences that help the encoder capture both long-range dependencies and local structures. Although the Morton curve preserves spatial locality, it fails to capture semantic continuity, while random ordering provides no structural guidance at all. In contrast, our greedy sorting enhances sequence stability and spatial representation, making it particularly effective for point cloud tasks with sequential architectures.

Table 2: Quantitative comparison of Sorting method

| Sorting method | ModelNet40 |
| --- | --- |
| | 2k P ↑ |
| greedy | 95.46 |
| morton curve | 94.93 |
| random | 94.77 |

Table 3: Comparison of different fine-tuning strategies on classification tasks

| Fine-tuning Strategy | Fine-tuning mode | ModelNet40 | ScanObjectNN |
|---|---|---|---|
| | | 2k P ↑ | PB_T50_RS ↑ |
| Baseline + Gate Adapter | fully finetuned | 95.46 | 94.06 |
| Baseline | fully finetuned | 95.01 | 93.90 |
| Adapter | PEFT | 95.01 | 92.57 |
| Adapter + TFTS | PEFT | 95.01 | 92.44 |

### 4.4.2 FINE-TUNING STRATEGY

As shown in Table 3, we analyze the impact of fine-tuning strategies on classification performance across ModelNet40 and ScanObjectNN to evaluate the effectiveness of the proposed *Gate Adapter*. Compared to conventional *Full Fine-tuning* (without Gate Adapter), our Gate Adapter approach consistently achieves higher accuracy, 95.46% on ModelNet40 and 94.06% on ScanObjectNN. These results highlight its dynamic fusion mechanism, which adaptively modulates feature transfer between pretraining and target domains to stabilize representation learning. In contrast, *Adapter* and *Adapter + TFTS (Task-specific Feature Scaling)* strategies, though parameter-efficient, suffer notable performance degradation on ScanObjectNN (92.57% and 92.44%), suggesting that insufficient structural adaptation limits generalization under noisy, real-world conditions. By using only a learnable gate without freezing the backbone, our method strikes a strong balance between generalization and stability. Overall, these findings validate the Gate Adapter as a practical strategy that retains the flexibility of full fine-tuning while enabling dynamic adaptability to diverse data distributions.

### 4.4.3 KH-NORM POOLING VARIANTS

Table 4: Quantitative comparison of KH norm

| KH norm | ModelNet40 |
|---|---|
| | 2k P ↑ |
| KNN+hybrid | 95.46 |
| KNN+Featpool | 95.09 |
| KNN+Distpool | 94.93 |
| KNN+Dirpool | 94.85 |

As shown in Table 4, we investigate the impact of different pooling mechanisms in KH-Norm on classification performance to validate the role of multi-geometry aggregation in local feature learning. We decompose the aggregation module into three variants: *FeatPool*, *DistPool*, and *DirPool*, each capturing distinct geometric cues such as feature intensity, point-wise distance, and directional distribution. While all variants achieve stable results (up to 95.09% with KNN+FeatPool), the integrated version, *KNN+Hybrid*, achieves the best accuracy of 95.46% on ModelNet40, demonstrating the superiority of combining multiple geometric perspectives. These findings confirm that hybrid aggregation enriches local structure understanding, thereby improving stability, generalization, and overall recognition performance.

## 5 CONCLUSION

In this work, we introduce *PointESS*, a structure-aware, task-adaptive framework for point cloud representation learning that unifies pre-training and downstream adaptation. PointESS is the first to leverage a Mamba2-based architecture for point cloud tasks, achieving strong performance with a compact model size through tailored feature embeddings and positional encoding, greedy sequence ordering, geometry-aware normalization (KH-Norm), and a lightweight, task-specific adaptation module (Gate Adapter). On ModelNet40, PointESS achieves 95.5% accuracy, surpassing most existing methods, and shows competitive robustness on ScanObjectNN.

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
