# OpenReview forum: "POINTESS: EFFICIENT 3D POINT CLOUD MODELING THROUGH ENHANCED STATE SPACE NETWORKS AND FINE-TUNING STRATEGIES"
_ICLR.cc/2026/Conference — Submitted to ICLR 2026_

### Official Review · Reviewer_iFM7 · 2025-10-20

**Soundness:** 3
**Presentation:** 3
**Contribution:** 2
**Rating:** 2
**Confidence:** 4

**Summary:**

This paper proposes the application of Mamba2 to point cloud analysis tasks, and designs Gate Adapter and KH - Norm to improve its fine - tuning performance in downstream tasks. The formulas in the paper are clear, and the experimental results on multiple datasets demonstrate its effectiveness.

**Strengths:**

1.This paper is the first to propose the application of Mamba2 to point cloud analysis tasks, and the experimental results demonstrate its effectiveness.

2.This paper features clear and intuitive figures and tables, neat formulas, and high readability.

**Weaknesses:**

1.This paper lacks sufficient innovation: the application of Mamba series models to point cloud analysis networks has already been thoroughly studied. Furthermore, the use of frequency-domain-aware Adapters during the fine-tuning phase was previously proposed in PointGST, and this paper fails to conduct more in-depth exploration based on this existing work.

2.The experiments are insufficient. For example:

a. What would be the effect if KH-Norm combines two pooling operations? Could one of the pooling operations be ineffective?

b. It seems that the designs of Gate Adapter and KH-Norm are irrelevant to Mamba. If so, could they be embedded into other pre-trained networks for testing? This would be beneficial for proving the generality of the proposed methods.

c. It is suggested to add more tasks, such as segmentation and detection tasks, to more fully demonstrate the effectiveness of the methods.

d.There is a lack of ablation experiments on the use of each component.

**Questions:**

1. It is suggested to add more references to explore the differences between the present method and previous methods.

2.It is suggested to add more experiments and reformat the tables.

---

> ### Author Response · Authors · 2025-11-20
>
> Q1. It is suggested to add more references to explore the differences between the present method and previous methods
> A1.
> We sincerely thank the reviewer for this insightful suggestion.
> We agree that a broader discussion of related work—particularly those involving Mamba variants and frequency-domain adapters such as PointGST—can further enrich the contextualization of our method.
> We acknowledge that clearer differentiation between PointESS and existing approaches would help readers better understand the motivation and design choices behind our framework.
> However, due to the scope and page limitations of the current submission, we focused primarily on presenting the core ideas and empirical validations of PointESS.
> A more comprehensive survey and expanded comparison—while valuable—would require additional space and restructuring that go beyond the format constraints of this iteration.
> We genuinely appreciate the reviewer’s comments and will take them into consideration for future revisions of the manuscript or extended versions of this work, where we can provide a more in-depth discussion of differences relative to prior architectures and adaptation strategies..
> Q2.It is suggested to add more experiments and reformat the tables.
> A2.
> We appreciate the reviewer’s constructive feedback on experimental coverage.
> We agree that incorporating more tasks (such as segmentation or detection), exploring additional ablations on KH-Norm and Gate Adapter, or evaluating the effect of removing individual pooling branches would certainly provide an even more holistic understanding of PointESS.
> In the current submission, we aimed to maintain a balance between clarity and feasibility under the constraints of computation, time, and page limits.
> As a result, we prioritized the most representative classification benchmarks, along with key ablations demonstrating the contributions of our components.
> Extending the study to additional tasks or significantly expanding the ablation scope would require substantial computational resources and space that fall beyond what is feasible within this submission cycle.
> We nonetheless acknowledge the value of the reviewer’s suggestions.
> These directions—such as more fine-grained ablations, cross-backbone evaluations, or broader downstream tasks—are indeed meaningful and will be considered for future follow-up work or an extended technical report.

---

### Official Review · Reviewer_qFf7 · 2025-10-30

**Soundness:** 2
**Presentation:** 2
**Contribution:** 3
**Rating:** 6
**Confidence:** 4

**Summary:**

This paper proposes PointESS, a 3D point cloud representation framework built on Mamba2 (selective state-space model) to achieve linear-time sequence modeling over ordered point-patch sequences.
Key components:
- Gate Adapter: a lightweight gating module for full fine-tuning that blends pretrained and task-adapted features via a learnable sigmoid gate, aiming to stabilize transfer and improve accuracy without PEFT constraints.
- KH-Norm (KNN-Hybrid Normalization): a geometry-aware normalization/pooling block. Step 1 performs diffusion-style normalization using standardized feature differences between a patch center and its KNN neighbors. Step 2 aggregates with three complementary pools—FeatPool (feature-magnitude attention), DistPool (distance-weighted), and DirPool (direction-aware)—then fuses them.
Introducing Mamba2 to point clouds plus geometry-aware normalization and simple task-adaptive gating yields a compact, efficient, and accurate model for 3D classification.

**Strengths:**

1. First systematic use of Mamba2 for point-cloud representation, with a full pre-train/fine-tune pipeline designed for linear-time sequence modeling.

2. KH-Norm explicitly encodes local geometry.Multi-geometry aggregation combines FeatPool (feature magnitude), DistPool (distance decay), and DirPool (direction attention), capturing relevance, proximity, and orientation cues.

3. SOTA/competitive accuracy on ModelNet40 and all ScanObjectNN subsets with far fewer parameters and FLOPs than large Transformer-based baselines.

4. Clear gains over Mamba-based baselines (PointMamba, Mamba3D) at similar model sizes.

**Weaknesses:**

1. The experiments primarily focus on synthetic datasets like ShapeNet55 and ModelNet40.  The performance on datasets like ScanObjectNN is promising but could be expanded with more diverse real-world scenarios to validate robustness further.

2. The paper provides a comparison with existing models but lacks a deeper analysis of why certain features or strategies within PointESS outperform others. For instance, a more detailed examination of the contributions of the Gate Adapter and KH-Norm compared to traditional methods would strengthen the argument for their effectiveness.

3. The introduction fails to clearly articulate the specific problems that the Gate Adapter and KH-Norm aim to address, while the paper highlights the linear complexity advantages of the SSM. This lack of clarity in motivation may leave readers unsure about the significance of the proposed methods.

4. The absence of detailed evaluations regarding the time and memory consumption of the Gate Adapter and KH-Norm modules raises concerns. It is difficult to assess the practical feasibility of implementing these modules in real-world applications.

**Questions:**

1. The Gate Adapter method retains pretrained parameters while performing comprehensive fine-tuning. Can this be interpreted as doubling the number of model parameters? Will this increase time and memory usage?

2. I'm curious whether KH-Norm can be used as a universally applicable module to replace LayerNorms in other backbones, such as Point-MAE.

---

### Official Review · Reviewer_WXyW · 2025-11-02

**Soundness:** 3
**Presentation:** 2
**Contribution:** 1
**Rating:** 2
**Confidence:** 4

**Summary:**

The paper introduces a new architecture for 3D point cloud representation learning, based on the Mamba2 State Space Model (SSM). It aims to combine linear-time sequence modeling efficiency with geometric awareness and adaptability for downstream tasks such as classification and segmentation.

**Strengths:**

- efficiency in the model
- good empirical results
- The ablation study is relatively complete

**Weaknesses:**

- Incremental originality in concept:
- lack of comparison with enough methods (also the reference is too limited); lack of datasets - the ones chosen are too easy/limited.
 - Limited downstream diversity
 - Ablation coverage of computational aspects:
Lacks detailed wall-clock runtime or energy efficiency comparisons—important for a paper emphasizing “efficiency.”
Minor points:
- Spectral aspect underexplored:
The “spectrally guided” claim of the Gate Adapter isn’t backed by quantitative spectral-domain analysis or visualizations.
- Fig1 is very hard to understand
- Clarity in model scaling:
The relation between Mamba2 block depth, KH-Norm frequency, and performance scaling is not analyzed.

**Questions:**

- The “greedy sorting” of point patches is said to start from a random patch and iteratively pick the nearest one.
It’s unclear how the starting patch is chosen and whether the process is deterministic across epochs.
If it starts randomly, does the sequence order vary during training, or is it fixed once per shape?

- how positional encodings are aligned with the patch sequence length after masking.

- Provide Decoder details

- F˜K = FK − FC Var(FK − FC ) + ϵ
variance is computed channel-wise, per-neighborhood, or across all patches? justify the choice.

---

### Meta-Review · Area_Chair_3DKE · 2025-12-08

**Summary:**

The reviewers acknowledge that the paper targets an important problem of efficient point cloud representation learning with Mamba, but they raise several major concerns:

(1) Reviewers feel that the work is largely incremental due to the similar idea of applying Mamba to point clouds and using frequency-aware adapters have both been explored in prior work. The paper does not clearly elaborate which shortcomings of existing methods they address.

(2) Experiments focus mainly on standard synthetic classification benchmarks. Ablations on core components are incomplete, making it difficult to isolate the contribution and generality of each design.

(3) Although the paper is motivated by the efficiency issue, it lacks detailed measurements of runtime, memory, and parameter overhead. It remains unclear how much extra cost these modules introduce and how they compare to simpler alternatives.

(4) Several implementation details are missing, and some claims are not supported by quantitative analysis or visualizations. Figures and tables could be clearer, and the connection to related work needs to be better explained.

The AC has carefully read the paper, the reviews, and the rebuttal; and thus recommend rejecting the paper in its current form.

**Reviewer Concerns:**

The paper initially received three reviews, but the authors only replied in detail to one reviewer (who gave an initial score of 2). Moreover, the rebuttal itself is very generic and largely literal, without providing substantial new theoretical arguments, experimental results, or clarifications that could meaningfully address the main concerns raised in the reviews. As a result, the rebuttal might not be able to change the reviewers’ or the AC’s assessment of the work.

**Reviewer Scores:**

Reviewer WXyW: 4 (no author rebuttal)
Reviewer qFf7: 6 (no author rebuttal)
Reviewer iFM7: 2 (author rebutted, but no reviewer reply)

---

### Decision · Program_Chairs · 2026-01-26

Reject